# TMPE Derived from Saffron Natural Monoterpene as Cytotoxic and Multidrug Resistance Reversing Agent in Colon Cancer Cells

**DOI:** 10.3390/ijms21207529

**Published:** 2020-10-13

**Authors:** Kamila Środa-Pomianek, Anna Palko-Łabuz, Andrzej Poła, Mirosława Ferens-Sieczkowska, Olga Wesołowska, Agata Kozioł

**Affiliations:** 1Department of Biophysics and Neurobiology, Wroclaw Medical University, ul. Chalubinskiego 3, 50-368 Wroclaw, Poland; kamila.sroda-pomianek@umed.wroc.pl (K.Ś.-P.); anna.palko-labuz@umed.wroc.pl (A.P.-Ł.); andrzej.pola@umed.wroc.pl (A.P.); 2Department of Chemistry and Immunochemistry, Wroclaw Medical University, ul. M. Skłodowskiej-Curie 48/50, 50-369 Wrocław, Poland; miroslawa.ferens-sieczkowska@umed.wroc.pl (M.F.-S.); agata.koziol@umed.wroc.pl (A.K.)

**Keywords:** monoterpene, saffron, β-cyclocitral, anticancer activity, multidrug resistance (MDR) reversal, ABCB1 transporter (P-glycoprotein), colon cancer

## Abstract

Terpenes constitute one of the largest groups of natural products. They exhibit a wide range of biological activities including antioxidant, anticancer, and drug resistance modulating properties. Saffron extract and its terpene constituents have been demonstrated to be cytotoxic against various types of cancer cells, including breast, liver, lung, pancreatic, and colorectal cancer. In the present work, we have studied anticancer properties of TMPE, a newly synthesized monoterpene derivative of β-cyclocitral—the main volatile produced by the stigmas of unripe crocuses. TMPE presented selective cytotoxic activity to doxorubicin-resistant colon cancer cells and was identified to be an effective MDR modulator in doxorubicin-resistant cancer cells. Synergy between this derivative and doxorubicin was observed. Most probably, TMPE inhibited transport activity of ABCB1 protein without affecting its expression level. Analysis of TMPE physicochemical parameters suggested it was not likely to be transported by ABCB1. Molecular modeling showed TMPE being more reactive molecule than the parental compound—β-cyclocitral. Analysis of electrostatic potential maps of both compounds prompted us to hypothesize that reduced reactivity as well as susceptibility to electrophilic attack were related to the lower general toxicity of β-cyclocitral. All of the above pointed to TMPE as an interesting candidate molecule for MDR reversal in cancer cells.

## 1. Introduction

Plants have lived on Earth for more than 400 million years, and the group survived and developed successfully in spite of being incessantly challenged by other organisms. Since plants have no ability to move and they do not possess immune system like animals, they have to rely on other means to protect themselves from herbivores and microbes. To accomplish this task, plants synthesize and store the vast group of structurally diverse compounds called secondary metabolites. The total number of secondary metabolites that have evolved in plants is estimated to exceed 200,000 compounds [1]. Their main function is to defend plants from herbivores and from bacterial, viral, and fungal infections. Secondary metabolites may also attract pollinating animals and serve as signaling molecules, antioxidants, and UV protectants [2]. The structural diversity of secondary metabolites is enormous, including alkaloids, non-protein amino acids, peptides, glycosides, lectins, terpenoids, phenolic compounds, and many others [2].

Terpenes constitute one of the largest groups of natural products; there have been more than 25,000 of these compounds reported [3]. Among ca. 55,000 of secondary metabolites identified in higher plants, terpenes constitute 16,000 entities, ca. 3000 of them being monoterpenes [2]. Chemically, terpenes are built from isoprene units (C5), the number of which decide on the type of terpene: monoterpenes (C10), sesquiterpenes (C15), diterpenes (C20), triterpenes (C30), tetraterpenes (C40), and polyterpenes. The majority of terpenes are lipophilic. Therefore, they are likely to interact with biological membranes and membrane proteins in an unspecific manner [1]. Terpenes exhibit a wide range of biological activities including antioxidant (reviewed in [4]), anticancer (reviewed in [5]), and drug resistance modulating activities (reviewed in [6]).

Due to their richness in secondary metabolites that are usually low molecular weight compounds with a wide array of biological activities, plants commonly have been and are used in traditional medicine. Plant-derived compounds also frequently constitute lead compounds for further chemical synthesis of derivatives of more beneficial pharmacological activity. Dried and powdered stigma of crocus (*Crocus sativus* L.) yields the yellow powder called saffron that is one of the most expensive spices in the world. Traditional medical use of saffron includes its application as diaphoretic, eupeptic, tranquilizer, expectorant, and abortifacient [7]. It was used to treat hepatic disorders, flatulence, spasms, vomiting, dental pain, insomnia, depression, seizures, cognitive disorders, and asthma. Nowadays, saffron is mainly applied as a food flavoring and coloring agent, but there are many reports emphasizing the potential health promoting activities of saffron constituents. Saffron extract and its constituents have been demonstrated to be cytotoxic against various types of cancer cells, including skin, breast, liver, lung, cervical, pancreatic, and colorectal cancer as well as leukemia (reviewed in [7,8,9]). Saffron extract was observed to inhibit nucleic acids synthesis, to inhibit cellular proliferation, to induce apoptosis, and to interfere with cellular antioxidant systems [7,8]. Saffron extract was also likely to be beneficial in alleviating symptoms of cardiovascular diseases, insulin resistance, depression, premenstrual syndrome, insomnia, and anxiety (reviewed in [10]). Among more than 150 compounds identified in saffron extract, the four most important, biologically active secondary metabolites have been identified. There are carotenoids, crocin and crocetin, and the monoterpene aldehydes, picrocrocin and safranal [7]. Safranal is the major constituent of saffron’s volatile oil responsible for its aroma. Another monoterpene in saffron is β-cyclocitral that differs from safranal by the absence of one double bond in the cyclohexane ring. β-Cyclocitral was identified to be the main volatile produced by the stigmas of unripe crocuses [11].

In the present work, we have studied anticancer properties of TMPE, a newly synthesized derivative of β-cyclocitral in colon cancer cells. Doxorubicin-sensitive (LoVo and HT-29) as well as doxorubicin-resistant (LoVo/Dx and HT29/Dx) colorectal cancer cell lines have been employed—the lines that differ in respect of the progress of the disease, LoVo being derived from more advanced tumor than HT-29. Therefore, in the present study, LoVo and HT-29 cells were employed as a model system for primary and more advanced cancer. Both doxorubicin-resistant LoVo/Dx and HT29/Dx cell lines are characterized by overexpression of ABCB1 protein (MDR1, P-glycoprotein), the main multispecific transporter involved in multidrug resistance (MDR) of cancer cells. MDR is a cellular phenomenon in which cancer cells protect themselves from cytostatic drugs mainly by pumping them out of the cell, thus reducing drug concentration below the killing threshold (reviewed in [12,13]). Among various MDR mechanisms, the overexpression of ATP-fueled ABC transporters that are able to get rid of various structurally unrelated drugs out of a cell seems to prevail.

## 2. Results

### 2.1. Synthesis of the Compound

The newly synthesized β-cyclocitral derivative was identified as ethyl 3-(2,6,6-trimethylcyclohex-1-en-1-yl) prop-2-enoate (TMPE) by LC-MS, ^1^H NMR, and ^13^C NMR. The chemical structures of the studied compounds are presented in Figure 1.

### 2.2. Cytotoxic Activity of β-Cyclocitral and its Derivative TMPE against Colon Cancer Cells

The effect exerted by β-cyclocitral and its derivative TMPE on cancer cell growth was studied in sensitive (LoVo, HT29) and resistant to doxorubicin colorectal adenocarcinoma cells (LoVo/Dx, HT29/Dx). The results indicated that β-cyclocitral in concentration up to 35 µM exerted virtually no inhibitory effect on the survival of any of four cancer cell lines (Figure 2A). In contrast, TMPE inhibited growth of resistant cells (Figure 2B), having no significant influence on sensitive cells. Both in LoVo/Dx and HT29/Dx, the cytotoxic effect of TMPE was concentration-dependent in concentrations up to 10 µM. Further increasing of monoterpene concentration did not change the extent of its cytotoxic activity. TMPE turned out to be more toxic to doxorubicin resistant LoVo/Dx and HT29/Dx cells than to sensitive LoVo and HT29 cells. This result prompted us to investigate TMPE as a candidate MDR modulating agent.

### 2.3. Chemosensitizing Effects

#### 2.3.1. Change in Doxorubicin Cytotoxicity in the Resistant Colon Cancer Cells

The compounds of interest were studied with regard to their ability to increase doxorubicin cytotoxicity in HT29/Dx and LoVo/Dx cells. LoVo/Dx and HT29/Dx cells are partially doxorubicin-resistant. Doxorubicin IC_50_ values obtained in HT29 and HT29/Dx cells were 1 ± 0.4 µM and 65 ± 3.7 µM, respectively. In our former study, the IC_50_ value for doxorubicin of 4 ± 0.9 µM for LoVo and 30 ± 3.7 µM for LoVo/Dx cells was reported [14]. Survival rate of resistant cells treated with doxorubicin and the studied monoterpenes was compared to survival rate of the cells treated with doxorubicin alone. Doxorubicin alone exerted cytotoxic effect that was concentration-dependent in both studied cell lines. Cytotoxicity of the drug was, however, limited, and slightly more than 50% of LoVo/Dx cells and ca. 40% of HT29/Dx cells survived the treatment with the highest concentration of doxorubicin used. At the same concentration of doxorubicin, the survival rates of LoVo and HT29 cells were ca. 10% less as compared to their resistant counterparts. In experiments in which combinations of the drug and monoterpene were used, β-cyclocitral and TMPE were applied at the concentration of 5 µM. In the presence of β-cyclocitral and TMPE, no changes in the cytotoxicity of doxorubicin against LoVo and HT29 cells were observed. On the other hand, the chemosensitizing effect of TMPE but not β-cyclocitral was observed in LoVo/Dx and HT29/Dx cells (Figure 3). TMPE has been identified as an effective MDR modulator since it increased doxorubicin cytotoxicity in both doxorubicin-resistant cell lines. In the presence of this derivative, the doxorubicin-mediated inhibition of LoVo/Dx cells growth was increased by about 40% and the observed effect was statistically significant in doxorubicin concentrations above 10 μM. The observed inhibitory effect of TMPE was even more pronounced in HT29/Dx cells than in the LoVo/Dx cell line and was statistically significant in doxorubicin concentrations above 25 μM.

When concentration and effect data obtained from SRB assay in LoVo/Dx and HT29/Dx cells were subjected to CompuSyn analysis, the synergy was observed between doxorubicin and TMPE but not between doxorubicin and β-cyclocitral (Table 1). The values of combination index obtained for doxorubicin combined with TMPE were below 1 in both cell lines.

#### 2.3.2. Increase in Doxorubicin Accumulation in Resistant Colon Cancer Cells

Additionally, the impact of β-cyclocitral and TMPE at 5 μM concentration on the accumulation of doxorubicin in human colon adenocarcinoma cells was investigated. In the presence of doxorubicin alone, intracellular accumulation of the drug was significantly higher in the sensitive cells (both LoVo and HT29) cells than in the appropriate resistant sublines LoVo/Dx and HT29/Dx (Figure 4). LoVo/Dx cells accumulated ca. seven times less doxorubicin than LoVo cells, while a ninefold decrease in drug accumulation was observed in the cases of the other studied pair of colon cancer cell lines. The addition of TMPE resulted in a significant increase of doxorubicin accumulation in both types of resistant cells. On the other hand, β-cyclocitral did not raise the accumulation of the anticancer drug significantly.

#### 2.3.3. Increase in Rhodamine 123 Accumulation in Colon Cancer Cells

One of the mechanisms of MDR reversal can be the direct inhibition of ABCB1 transporter. To confirm the putative role of TMPE as an MDR modulator, a flow cytometric test was applied. Rhodamine 123 (Rho123), a known substrate of ABCB1, was used. TMPE was demonstrated to inhibit ABCB1 transport activity in HT29/DX cells as illustrated by the increase of FIR values when the concentration of the drug was raised (Figure 5). FIR values increased along with concentration of TMPE up to 50 μM. Instead, β-cyclocitral did not increase accumulation of Rho123 in these cells. When the activity of TMPE was compared with verapamil, the known inhibitor of ABCB1 transporter, it was found that FIR values of TMPE were higher in case of LoVo/Dx cells and comparable to verapamil in HT29/Dx cells (Figure 5).

#### 2.3.4. ABCB1 Expression in LoVo/Dx and HT29/Dx Cells

To check whether the observed TMPE-induced resistance reversal effect was due solely to the inhibition of ABCB1 activity by this derivative, the effect of TMPE on ABCB1 expression was investigated. LoVo/Dx and HT29/Dx cells were cultivated in the presence of either β-cyclocitral or TMPE at 5 µM concentration for 48 h. It was found that neither β-cyclocitral nor TMPE changed the level of *ABCB1* mRNA (Figure 6). Therefore, a Western blot experiment was conducted to check whether the treatment of LoVo/Dx and HT29/Dx cells with monoterpenes affected the expression of ABCB1 protein. Again, it was observed that ABCB1 protein level remained unchanged either in the presence of β-cyclocitral or TMPE (Figure 7).

### 2.4. Molecular Calculations

In order to characterize the chemical properties and putative biological activity of the β-cyclocitral and its derivative, TMPE, physical parameters describing the structure and reactivity of the compounds were calculated. The SPARTAN’18 software package was employed, and the equations presented in [15] were used. The values of the calculated molecular descriptors are presented in Table 2.

HOMO (highest occupied molecular orbital) and LUMO (lowest unoccupied molecular orbital) energy levels define how the molecule shares its valence electrons in the occupied molecular orbitals and donates (or accepts) electrons to a ligand. The energy gap (∆E) indicates the energy difference between HOMO and LUMO (Figure 8). In β-cyclocitral molecule HOMO orbitals were localized mainly at the double bond between carbon atoms in the hexagonal ring, whereas LUMO orbitals were localized near the single bond connecting the ring with the aldehyde group. HOMO orbitals in TMPE were mainly present near double C-C bonds (both in the ring and the chain) and LUMO orbitals mainly in the side chain region of the molecule.

TMPE occurred to be the more reactive compound than the parent β-cyclocitral since the energy gap for TMPE had a lower value. This was in agreement with β-cyclocitral having higher hardness value since this parameter measures the resistance of a molecule to change of the electron distribution. On the other hand, global electrophilicity index was higher for TMPE that suggested that this derivative accepted electron more easily than the parent compound. Looking at the calculated values of dipole moment and logP, one can notice that β-cyclocitral was more polar and TMPE more hydrophobic of the two compounds.

Figure 9 represents electrostatic potential maps, also called molecular electrical potential surfaces, of β-cyclocitral and TMPE. These maps allow for visualization of variably charged regions of a molecule. The blue color represents the positive region that is prone to nucleophilic attack, while the red color represents the negative region, therefore, is prone to electrophilic attack. The values of maximum positive electrostatic potential (blue region in Figure 9) were 7.13 kcal/mol and 5.31 kcal/mol for β-cyclocitral and TMPE, respectively. The maximum negative electrostatic potential (red region in Figure 9) was located at oxygen atoms in both molecules. The values of negative potential were −14.06 kcal/mol and −12.98 kcal/mol for β-cyclocitral and TMPE, respectively. For both compounds, the absolute value of the maximum negative potential was bigger than the value of positive potential, so both compounds were likely to act as nucleophiles. The greater value of negative maximum potential for β-cyclocitral suggested this compound to be a better nucleophile than TMPE.

### 2.5. Prediction of Toxicological and Physicochemical Properties

Since TMPE has been identified as an effective MDR modulator in colon cancer cells, an attempt to predict its toxicity risks and some significant physicochemical properties was made using OSIRIS-Property-Explorer. The results are summarized in Table 3. Similarly, as in the case of SPARTAN’18-calculated parameters, TMPE was identified to be more hydrophobic and slightly less soluble in water than β-cyclocitral. Due to the longer side chain substituted to the cyclic ring, TMPE had higher molecular weight and bigger topological polar surface area than β-cyclocitral. On the other hand, β-cyclocitral showed higher drug-likeness. Drug Score is an OSIRIS-calculated parameter that combines all previously listed descriptors and allows for prediction of the molecule’s potential to qualify as a drug. TMPE turned out to be slightly better than β-cyclocitral in this respect. The toxicity risk predictor showed that TMPE was not likely to be mutagenic, tumorigenic, irritant, or to affect reproduction. On the other hand, the risk of being irritant was recognized for β-cyclocitral.

## 3. Discussion

Anticancer activity of β-cyclocitral and its derivative, TMPE, was investigated in the two colon cancer cell lines LoVo and HT29, as well as in their doxorubicin-resistant sublines LoVo/Dx and HT29/Dx. The lines are derived from the same type of cancer—colorectal adenocarcinoma—but they differ in respect of the progress of the disease. The most common tumor classification system is TNM (tumors/nodes/metastases) in which increasing numbers are used to describe tumor progression in three categories: T denotes the degree of invasion of the intestinal wall; N gives information about the degree of lymphatic node involvement; and M describes the degree of metastasis. A shorter format of the TNM tumor staging is also used in which numbers I, II, III, and IV are attributed to TNM values grouped by prognosis (higher number means more advanced disease and worse prognosis). HT29 cell line was derived from the primary colorectal tumor [16] (stage I/II), so it was used as a cellular model of the beginning of neoplastic process. LoVo cell line was derived from metastatic site and represents Dukes’ type C grade of cancer [17] (stage III); therefore, in the present study, it modeled more advanced neoplastic process. It is well known that the epithelial to mesenchymal transition (EMT) is a mechanism that contributes to tumor progression, cancer cell invasion, and therapy resistance. During EMT, epithelial cells lose their polarity and cell–cell adhesion, instead gaining migratory and invasive properties. Many families of transcriptional regulators acting via various signaling pathways are engaged in the control of the EMT process [18]. Molecular changes that take place during EMT, e.g., during the processes of differentiation may also contribute to MDR development [19]. The sublines LoVo/Dx and HT29/Dx were obtained by continuous exposure to low concentration of doxorubicin in order to induce drug resistance [20,21]. Thus, the sublines HT29/Dx and LoVo/Dx constituted an in vitro model of resistant colon cells.

Initially, cytotoxicity of the studied monoterpenes was investigated. β-Cyclocitral in concentrations up to 35 µM was not cytotoxic to any type of cancer cells studied. TMPE, tested in the same concentration range, was found to inhibit cell growth only of LoVo/Dx and HT29/Dx cells. During our studies we usually observed that doxorubicin-resistant cells tended to be less vulnerable to other chemicals than their sensitive counterparts [22,23,24]; thus, the result was particularly important. The ability of various monoterpenes to inhibit cancer cell growth was previously demonstrated. Obtained IC_50_ values were dependent on structure of monoterpene molecule as well as on cancer cell line used. For linearly-shaped citral, IC_50_ values ranged from 10–15 µM in melanoma cell lines [25] to 60–80 µM in breast [26] and small cell lung cancer [27]. On the other hand, cyclic monoterpenes were usually characterized by IC_50_ values close to several hundred µM (e.g., β-elemene [28], safranal [29], carvacrol, and thymol [30]); however, one study reported IC_50_ values below 1 µM [31]. Similar to our results on TMPE, de Ines et al., in their study on cytotoxic activity of nine halogenated monoterpenes, observed that one of the compounds was selectively cytotoxic to SW480 human colon adenocarcinoma cell line that overexpressed ABCB1 transporter [32]. Additionally, it was noticed that gemcitabine-resistant breast cancer cells were less vulnerable to carveol, carvone, and eugenol than their sensitive counterparts but more vulnerable to carvacrol [31]. Selective cytotoxic activity of TMPE to doxorubicin-resistant colon cancer cells prompted us to study its potential as a putative MDR modulator.

It has been shown that TMPE was able to increase doxorubicin cytotoxicity in both LoVo/Dx and HT29/Dx cells, i.e., to reduce the resistance of cancer cells to the anticancer drug. The resistance to doxorubicin in LoVo/Dx cells is mainly due to the elevated expression of ABCB1 transporter in these cells as compared to their sensitive couterparts, LoVo [22,33]. ABCB1, but also other MDR-associated transporters, were also overexpressed in HT29/Dx cells [34]. Additionally, TMPE caused significant increase of accumulation of doxorubicin and ABCB1 substrate, rhodamine 123, inside the resistant colon cancer cells. These results suggested that TMPE inhibited transport activity of ABCB1 protein. It was also checked whether TMPE affected the expression of this transporter, but no effect was observed. Other monoterpenoids were also reported to interfere with ABCB1 function. Menthol and thymol caused the increase of doxorubicin cytotoxicity in human colorectal adenocarcinoma cells (Caco-2) and doxorubicin-resistant leukemia cells (CEM/ADR5000) [35]. Both compounds moderately inhibited ABCB1 transport function and decreased its expression in Caco-2 but not in CEM/ADR5000 cells [36]. Yoshida et al. observed that some monoterpenoids from *Zanthoxyli fructus* were inhibitors of ABCB1 protein [37] but they did not interfere with transport activity of ABCC2 and ABCBG2 [38]. Safranal was also found to affect ABCB1 and ABCG2 transporters, increasing their ATP-ase activity [39]. Potent MDR modulators have been identified among more complex terpenes such as jathropane [40] and lathyrane diterpenes [24,41].

Both in LoVo/Dx and HT29/Dx cells, a synergistic effect between doxorubicin and TMPE was found. Synergy between anticancer drugs and monoterpenes have been observed only in few combinations. Thymol and menthol [33] as well as linalool [42] exhibited synergism with doxorubicin in leukemia and colon cancer cells. The combination of carvacrol, thymol, carveol, carvone, eugenol, and isopulegol with either methotrexate or cis-platin resulted in synergy between terpenes and anticancer drugs [31]. Moreover, synergistic fungicidal activity against *Candida albicans* of geraniol combined with fluconazole was revealed [43] as well as synergy of thymol and eugenol co-administered with streptomycin against *Listeria monocytogenes* and *Salmonella typhimurium* [44]. The authors concluded that observed synergy was related to the ability of monoterpenes to inhibit MDR-associated transporters.

Molecular modeling demonstrated TMPE being more reactive molecule than β-cyclocitral judging from the value of energy gap between HOMO and LUMO orbitals. Analysis of electrostatic potential maps of both compounds pointed to β-cyclocitral being more nucleophilic (electron donating) and TMPE more electrophilic (electron withdrawing) compound. Therefore, it was hypothesized that reduced reactivity as well as susceptibility to electrophilic attack were related to the lower general toxicity of β-cyclocitral. Both SPARTAN’18 and OSIRIS calculations yielded higher logP value for TMPE than for β-cyclocitral. The presence of longer side chain in TMPE molecule resulted in an elevated hydrophobicity of this derivative. Regarding the molecular weight, logP value and the number of H-bond donors and acceptors, both compounds fulfilled Lipinski’s rule of five commonly used to judge chemical compounds as potential drug candidates [45]. Drug score parameter calculated by OSIRIS software is the combination of drug likeness, cLogP, logS, molecular weight, and toxicity risks into one handy value that could be used to rate a chemical’s potential to qualify as a drug. TMPE scored higher than β-cyclocitral in this respect, probably due to the high probability of irritant effects expected for the latter compound. On the other hand, when physicochemical parameters of TMPE were analyzed for compliance with the “rule of four” proposed by Didziapetris et al., to recognize ABCB1 substrates [46], it turned out that TMPE was not likely to be a substrate of this transporter.

In conclusion, TMPE was identified as an effective MDR modulator in doxorubicin-resistant colon cancer cells. Synergy between this derivative and doxorubicin was observed. It was demonstrated that TMPE reversed resistance of LoVo/Dx and HT29/Dx cells to doxorubicin and increased the accumulation of the drug within cancer cells. These results, together with a TMPE-induced increase in ABCB1 substrate rhodamine 123 accumulation, suggested that TMPE was able to interfere with the transport activity of ABCB1 protein. On the other hand, TMPE did not affect expression of this transporter. The elucidation of the detailed mechanism of ABCB1 inhibition by TMPE requires further studies. However, on the basis of the analysis of physicochemical parameters of TMPE, it was concluded it was not likely to be transported by ABCB1 itself. Therefore, the putative mechanism of reducing of ABCB1 activity by TMPE most probably is not purely competitive inhibition. Nevertheless, it would be possible that TMPE interacted with transporter protein at the site that was only partially overlapping or separate from the binding site for substrates. Another important observation was that in the studied concentration range TMPE was selectively cytotoxic only to resistant cancel cells. All of the above make TMPE to be an interesting candidate molecule for MDR reversal in cancer cells.

## 4. Materials and Methods

### 4.1. Chemicals

#### Synthesis

The substrate for the derivatization procedure was β-cyclocitral. The reaction mixture was prepared as follows: NaH was added to 10 mL of tetrahydrofuran to a final concentration of 0.019 M. In the same manner, a second reagent was prepared, 10 mL of tetrahydrofuran containing 0.036 M triethylacetate, which was then added dropwise to the sodium hydride solution with constant stirring. The mixture was incubated at room temperature for 15 min. 1.9 g of β-cyclocitral was dissolved in 20 mL of tetrahydrofuran, and then added dropwise to the previously prepared mixture. The reaction was carried out at room temperature for about 18 h, checking the appearance of the reaction product by TLC. After the presence of the product was confirmed, the reaction mixture was washed with an equal amount of water, and, after removal of the aqueous phase, the excess of unreacted substrates was removed by triple extraction with hexane. The final product was dehydrated by adding anhydrous MgSO_4_. The crude product was purified by flash chromatography on a silica gel (Biotage Sfär silica column, Biotage, Sweden) washed with 5:3 hexane/ethyl acetate.

Analytical Data for free base:

Formula: C_14_H_22_O_2_ (2): Oily substance, 51%, ^1^H NMR (CDCl_3_, 600, MHz) 1.03-1.06 (3H, m, 2CH_3_), 1.39-1.42 (3H, m, CH_3_), 1.60-1.63 (2H,m, CH_2_), 1.68-1.72 (3H, m, CH_3_), 1.95 (2H, t, J = 6.4 Hz, CH_2_), 2.05 (2H, m, CH_2_), 4.10-4.18 (2H, m, CH_2_), 5.94 (1H, s, CH), 7.29 (1H, s, COO). ^13^C NMR (150, CDCl_3_, MHz) 15.96 (CH_3_, C-15), 20.72 (CH_2_, C-5), 21.44 (CH_3_, C-12), 28.48 (CH_3_, C-13), 28.56 (CH_3_, C-14), 33.72 (CH_2_, C-4), 34.61 (C, C-6), 35.64 (CH_2_, C-3), 61.54 (CH_2_, C-10), 107.77 (CH, C-8), 124.06 (C, C-2), 134.70 (C, C-1), 148.51 (CH, C-7), 174.02 (C, C-9). HRMS m/z 223.1699 obtained for C_14_H_22_O_2_, 223.1698).

### 4.2. Cell Cultures

Human colon cancer cells HT29 and doxorubicin-resistant HT29/Dx cells were a kind gift from Prof. Chiara Riganti (Department of Genetics, Biology and Biochemistry, University of Torino, and Research Center on Experimental Medicine, Turin, Italy) and have been previously characterized [21]. The human colorectal adenocarcinoma cell line LoVo and its doxorubicin–resistant subline LoVo/Dx [17,20] were obtained from Institute of Immunology and Experimental Therapy of Polish Academy of Science (Wroclaw, Poland). Experimental procedures were carried out in log-phase of cell growth. The cells were cultured in F12 medium (Cytogen, Princeton, NJ, USA) supplemented with 10% fetal bovine serum (Gibco), 1% antibiotic antimycotic solution (Sigma, Taufkirchen, Germany) and 1% glutamine (Sigma, Taufkirchen, Germany).

The cell lines were incubated in a humidified atmosphere (5% CO_2_, 95% air) at 37 °C. Doxorubicin at concentration 100 ng/mL was added to the medium to maintain drug resistance of LoVo/Dx and HT29/Dx cells after each passage which was carried out twice a week. The adherent cancer cells were detached with non-enzymatic cell dissociation solution (Sigma, Taufkirchen, Germany). To determine density of cells for each experiment the EVE Automatic Cell Counter (NanoEnTek, Seoul, Korea) was used.

### 4.3. Cell Viability Assay

The sulforhodamine B (SRB) assay [47] was modified and used for the estimation the effects of the studied compounds alone and in combinations on cell growth. Cells (30,000/well) were seeded in 96-well flat-bottomed microtiter plates in 75 μL of medium and allowed to attach (60 min, 37 °C). Next, 75 μL of medium containing an appropriate concentration of the studied compounds was added to each well, except for the medium control wells. The culture plates were further incubated at 37 °C for 48 h. The further procedure was carried out as previously described [48]. Cytotoxicity of DMSO to the LoVo and LoVo/Dx cells was found to be negligible.

### 4.4. Polymerase Chain Reaction

To examine the ability of β-cyclocitral and TMPE to affect the expression level of ABCB1 gene semiquantitative RT-PCR method was used. Cells were seeded onto 6-well plates (37 °C for 24 h) and allowed to attach onto the plate surface. Subsequently, β -cyclocitral and TMPE were added to obtain 5 μM concentration. Cells were incubated in the presence of the studied compounds at 37 °C for 48 h. The primers for *ABCB1* and *β-GUS* (reference gene) were designed as previously described [14]. To determine their specificity and compare to the data from Genbank software Blast (www.ncbi.nlm.nih.gov) was used. All the sequences were synthesized in Institute of Biochemistry and Biophysics of Polish Academy of Science (Warsaw, Poland). Gel Documentation System KODAK MI v4.0.0. was used to visualize the separated fragments. The relative level of ABCB1 expression, normalized to the control, was estimated by detection of optical density of the bands on the electrophoregrams using Image J software.

### 4.5. Western Blot Analysis

Whole-cell extracts were directly solubilized in an ice cold sample buffer (1%Triton X-100, 50 mM Hepes, 150 mM NaCl, 1.5 mM MgCl_2_, 1 mM EGTA, 1 mM phenylmethylsulfonyl fluoride (PMSF),100 mM NaF, 10 mM sodium pyrophosphate, 10 µg/mL aprotinin, and10% glycerol, pH 7.4). Next, whole cell lysates were centrifuged (13,000× *g*, 10 min, 4 °C) and the supernatants were collected for analysis. The protein content was determined using the standard method of Bradford reaction [49]. Proteins were separated by SDS/PAGE (12% or 7% gels) and transferred onto polyvinylidene difluoride (PVDF) membranes. Next, Ponceau stain was performed. The blots were probed with antibodies diluted in TBS (25 mmol/l Tris/HCl, pH 7.6, and 150 mmol/l NaCl)/Tween (0.1%) with 5% defatted dried milk. The ABCB1 (C494, dilution 1:1000, Alexis) mouse monoclonal primary antibodies were used. The level of β-actin was also determined (mouse monoclonal anti-β-actin antibodies (C4), diluted 1:5000) as a reference protein. After incubation, the membranes were washed in TBS-T and incubated with the Pierce Rabbit Anti-Mouse IgG (HRP-conjugated) Secondary Antibody (dilution 1:1000, Thermo Scientific, Waltham, MA, USA) at 4 °C for 30 min. The membranes were washed with TBS-T and the proteins were visualized with Ponceau staining. The relative level of protein normalized to the control derived from non-treated cells was determined. Densitometric analysis of the bands was performed with ImageJ software.

### 4.6. Intracellular Accumulation of Doxorubicin

Intracellular doxorubicin accumulation was measured using a LS-5 spectrofluorimeter (Perkin-Elmer, Beaconsfield, UK) as described previously [50]. Excitation and emission wavelengths were 475 and 553 nm, respectively. Fluorescence was expressed in ng of doxorubicinox per mg of cellular protein with the use of the calibration curve prepared previously.

### 4.7. Accumulation of Rhodamine 123 in Cancer Cells

Flow cytometry was used to evaluate the effects of the studied compounds on the accumulation of rhodamine 123 by ABCB1 in cancer cells. To determinate rhodamine 123 accumulation, cells (3 × 10^5^ cells/mL) were incubated with the appropriate concentration of β-cyclocitral and TMPE (15 min, 37 °C). Next, rhodamine 123 (2 μM) was added and the cells were incubated for further 60 min at 37 °C. After centrifugation, the samples were washed twice with ice-cold PBS. Flow cytometric analysis of rhodamine 123 fluorescence via 530/30 nm band pass filter was carried out with the BD FACS Control, Canto II instrument (Becton Dickinson, Franklin Lakes, NJ, USA) equipped with a 488 nm argon laser. A total of 5000 events were acquired per sample and analyzed with the use of Cell Quest^®^ software (Becton Dickinson, Franklin Lakes, NJ, USA). On the basis of measured fluorescence values of studied and control (without modulator) samples, the fluorescence intensity ratio (FIR) was calculated according to the equation:(1)$$\frac{\left(FL_{LoVoDx\;or\;HT29/Dx\;treated})/(FL_{LoVoDx\;or\;HT29/Dx\;control}\right)}{\left(FL_{LoVo\;or\;HT29\;treated}\right)/\left(FL_{LoVo\;or\;HT29\;control}\right)}$$

### 4.8. Isobolographic Analysis

CompuSyn software (www.combosyn.com, ComboSyn, Inc., Paramus, NJ, USA) was used to calculate the combination index (CI) values according to the classic median-effect equation described by Chou and Martin [51]:(2)$$CI=\frac{{\left(D\right)}_{1}}{{\left(Dx\right)}_{1}}+\frac{{\left(D\right)}_{2}}{{\left(Dx\right)}_{2}}$$
where: (*Dx*)_1_ is the concentration of drug 1 alone that inhibits a system by x%, (*Dx*)_2_ is the concentration of drug 2 alone that inhibits a system by x%, and (*D*)_1_ + (*D*)_2_ are concentrations of drug 1 and 2 in combination that also inhibit a system by x%.

### 4.9. Molecular Calculations

Visualization of variably charged regions of studied compounds as well as all quantum calculations were performed using the SPARTAN’18 calculation package (Wavefunction, Inc., Irvine, CA, USA). Geometry of the molecule in aqueous phase of each investigated compound was optimized using the density functional theory (DFT) method with ωB97X-D exchange correlation potential (including empirical corrections for dispersive interactions) in connection with 6-311+G ** basic set. The optimized geometry of the given molecule was confirmed to be the real minimum by frequency analysis (no imaginary frequencies). The solvent effect on geometry and value of quantum mechanical parameters was assessed using the conductor-like polarizable continuum model (CPCM) method.

The toxicological risk and physicochemical properties of the studied molecules were obtained using the OSIRIS property explorer [52]. The risks of side effects, such as mutagenic, tumorigenic, irritant, and reproductive effects, as well as drug-relevant properties including cLogP, LogS (solubility), molecular weight, drug-likeness and overall drug-score were estimated.

### 4.10. Statistical Analysis

The values of all the measured parameters were presented as means from three independent experiments ± standard deviations (S.D.). The statistical significance was determined by Student’s *t*-test (0.05 as a threshold value).

## Figures and Tables

**Figure 1 ijms-21-07529-f001:**
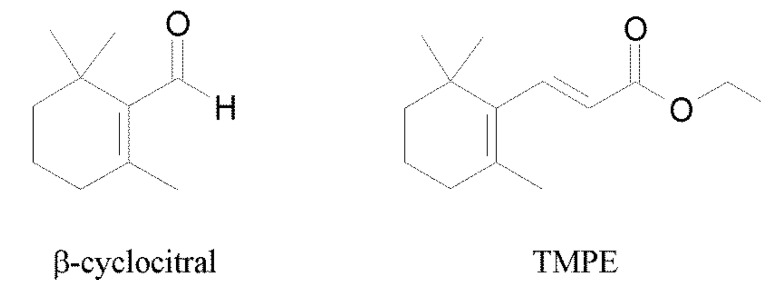
Chemical structures of β -cyclocitral and ethyl 3-(2,6,6-trimethylcyclohex-1-en-1-yl) prop-2-enoate (TMPE).

**Figure 2 ijms-21-07529-f002:**
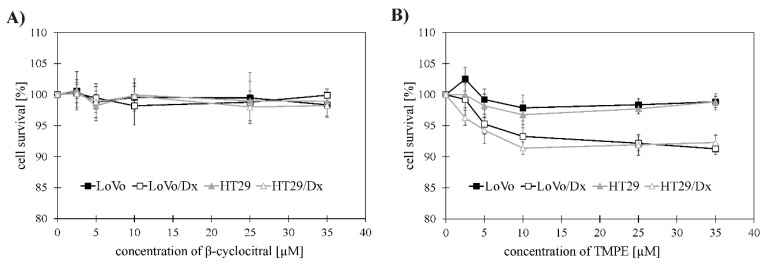
Cytotoxic activity of β-cyclocitral (**A**) and TMPE (**B**) against LoVo, LoVo/Dx, HT29, and HT29/Dx cells. The results were shown as mean + S.D. from three independent experiments. Cell survival at the concentrations of 2.5, 5, 10, 25, and 35 µM of the studied compounds were compared to the results obtained for the control (without the compound). The statistical significance was determined using Student’s *t*-test (*p* < 0.05). Statistical significance was found only for TMPE in HT29/Dx cells (for all concentrations tested) and in LoVo/Dx cells (for concentrations equal or greater than 5 µM). Doubling times were: 36.44 ± 5.18 h for LoVo, 27.07 ± 4.53 h for LoVo/Dx, 30.02 ± 5.89 h for HT29, and 21.51 ± 3.27 h for HT29/Dx cell line.

**Figure 3 ijms-21-07529-f003:**
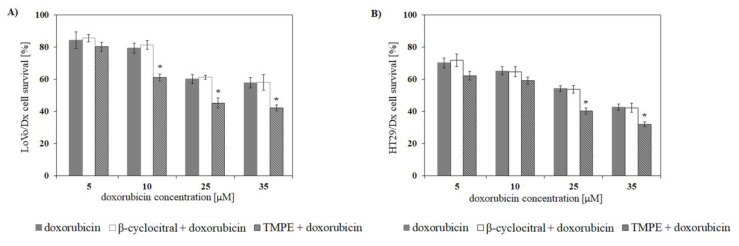
Changes in cytotoxicity of doxorubicin against LoVo/Dx (**A**) and HT29/Dx (**B**) cells in the presence of 5 µM of β-cyclocitral and TMPE. The results were shown as mean ± S.D. from three independent experiments. The statistical significance between cytotoxicity observed in the presence of doxorubicin alone and doxorubicin used together with monoterpene compounds was determined using Student’s *t*-test (* *p* < 0.05).

**Figure 4 ijms-21-07529-f004:**
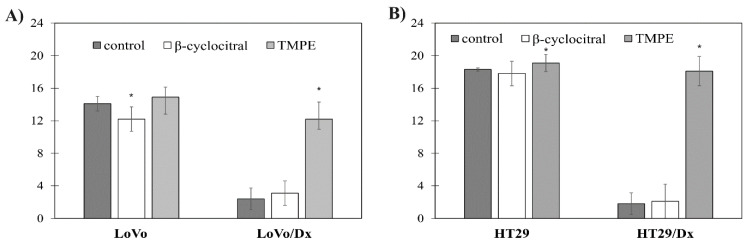
The effect of β-cyclocitral (5 µM) and TMPE (5 µM) on the intracellular accumulation of doxorubicin in cultures of LoVo and LoVo/Dx (**A**), as well as HT29 and HT29/Dx (**B**) cells. The results were shown as mean ± S.D. from three independent experiments. The statistical significance was determined using Student’s *t*-test (* *p* < 0.05).

**Figure 5 ijms-21-07529-f005:**
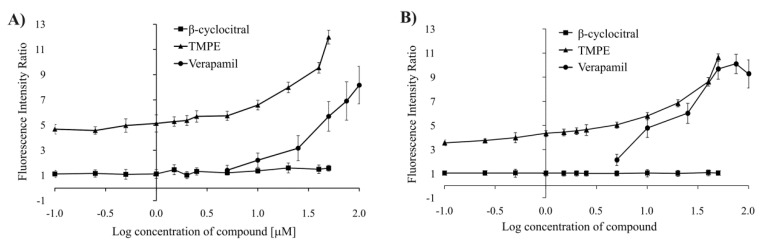
The influence of β-cyclocitral and TMPE on the intracellular accumulation of rhodamine 123 (Rho123) in LoVo/Dx (**A**) and HT29/Dx cells (**B**). Mean ± S.D. of three experiments are presented. The following FIR values were found to be statistically significant (*p* < 0.05) as compared to the control: TMPE in all concentrations in LoVo/Dx cells; TMPE in all concentrations in HT-29/Dx cells; β-cyclocitral at the highest concentration in LoVo/Dx cells; verapamil in all concentrations (apart from the lowest) in LoVo/Dx cells; verapamil in all concentrations in HT-29/Dx cells.

**Figure 6 ijms-21-07529-f006:**
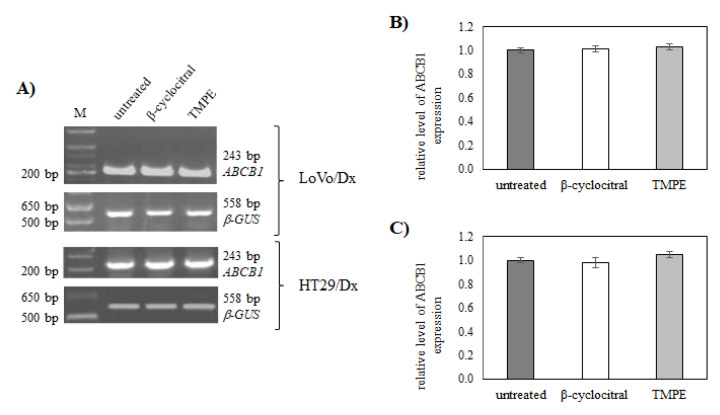
The influence of β-cyclocitral (5 µM) and TMPE (5 µM) on the *ABCB1* level in LoVo/Dx and HT29/Dx cells (**A**). The base pair lengths of the amplified products are indicated at the left side of the gel. β*-GUS* was used as a reference gene. The relative level of *ABCB1* expression in LoVo/Dx (**B**) and HT29/Dx cells (**C**) normalized to the control derived from non-treated cells. The results of three experiments (mean ± S.D.) are presented. The statistically significant differences from the untreated controls were determined using Student’s *t*-test (*p* < 0.05).

**Figure 7 ijms-21-07529-f007:**
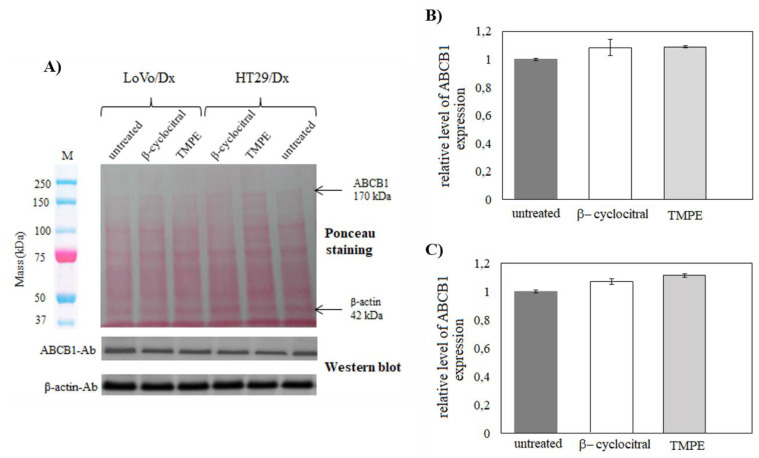
The influence of β-cyclocitral (5 µM) and TMPE (5 µM) on the ABCB1 protein expression in LoVo/Dx and HT29/Dx cells (**A**). Samples were analyzed by Western blot, stained with Ponceau, and probed with anti-ABCB1 and anti-β-actin antibodies. The respective molecular weights are indicated at the right side of the gel. The relative level of ABCB1 expression in LoVo/Dx (**B**) and HT29/Dx cells (**C**) normalized to the control derived from non-treated cells. The results of three experiments (mean ± S.D.) are presented. The statistically significant differences from the untreated controls were determined using Student’s *t*-test (*p* < 0.05).

**Figure 8 ijms-21-07529-f008:**
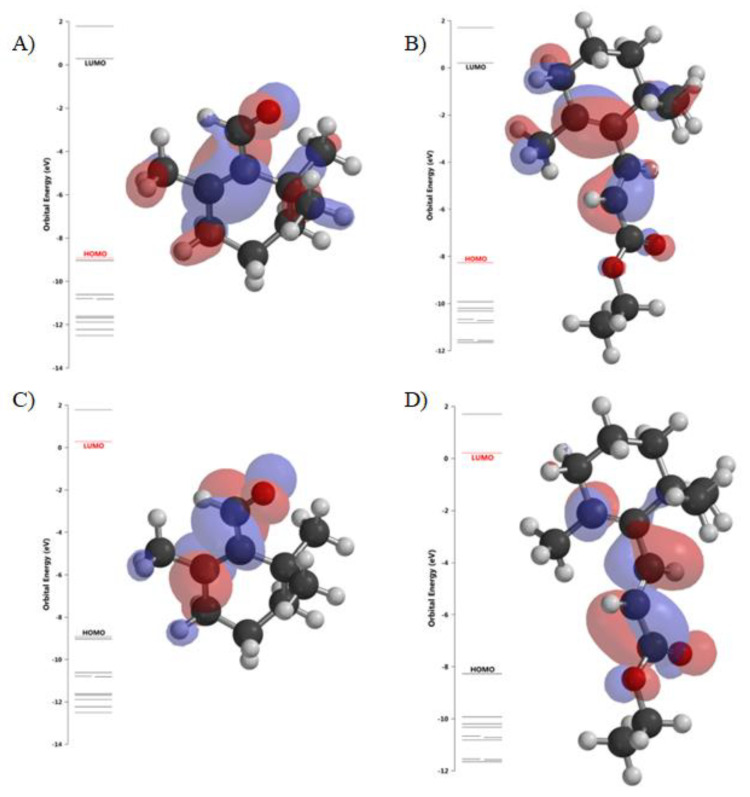
HOMO (**A**,**B**) and LUMO (**C**,**D**) orbitals for β-cyclocitral (**A**,**C**) and TMPE (**B**,**D**) in aqueous solution. The images were obtained using SPARTAN software.

**Figure 9 ijms-21-07529-f009:**
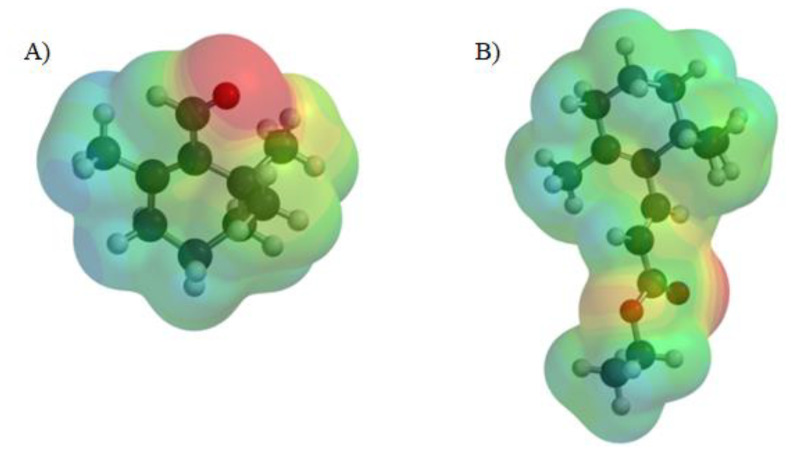
Electrostatic potential maps for β-cyclocitral (**A**) and TMPE (**B**) in aqueous solution. The blue color represents the positive region and the red color represents the negative region. The images were obtained using SPARTAN software.

**Table 1 ijms-21-07529-t001:** Combination β-cyclocitral and its derivative TMPE with doxorubicin (Dox) against LoVo/Dx (**A**) and HT29/Dx (**B**) cell growth.

**(A)**				
**Concentration (µM)**			**Ratio**	**Combination Index**
**Dox**	**β-Cyclocitral**	**TMPE**		
10	5		2:1	1.11
10		5	2:1	0.63
**(B)**				
**Concentration (µM)**			**Ratio**	**Combination Index**
**Dox**	**β-Cyclocitral**	**TMPE**		
10	5		2:1	1.201
10		5	2:1	0.852

Concentration and effect data were obtained from the SRB assay (mean values of three experiments) and analyzed by CompuSyn software. CI values were calculated by CompuSyn software. CI = 1 indicates additive effect, CI < 1—synergism, and CI > 1—antagonism.

**Table 2 ijms-21-07529-t002:** Theoretical parameters obtained for studied compounds (in water) using the SPARTAN’18 calculation package.

Parameter	β-Cyclocitral Aqua	TMPE Aqua
E_LUMO_ (eV)	0.28	0.42
E_HOMO_ (eV)	−8.92	−8.35
Energy gap (eV)	9.20	8.48
Chemical hardness (eV)	4.60	4.24
LogP	2.14	3.52
Dipole moment (D)	6.06	3.15
Global electrophilicity index (eV)	1.793	1.915

**Table 3 ijms-21-07529-t003:** OSIRIS toxicological and physicochemical predicted properties.

Property		Compound	
		β-Cyclocitral	TMPE
Physico-chemical properties	cLogP	2.118	3.472
	Solubility (LogS)	−2.214	−2.721
	Molecular weight	152	222
	TPSA	17.07	26.30
	Druglikeness	−7.062	−11.216
	Drug Score	0.281	0.426
Toxicity risks	Mutagenic	N	N
	Tumorigenic	N	N
	Irritant	H	N
	Reproductive effect	N	N

N—no risk; H—high risk. TPSA—topological polar surface area.

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
