# Peer review of "TMPE Derived from Saffron Natural Monoterpene as Cytotoxic and Multidrug Resistance Reversing Agent in Colon Cancer Cells"

_ijms, 2020, doi:10.3390/ijms21207529_

Round 1

Reviewer 1 Report

The paper “TMPE derived from saffron natural monoterpene as cytotoxic and multidrug resistance reversing agent in colon cancer cells” by Środa-Pomianek and co-workers investigates the anticancer properties of β-cyclocitral and TMPE against colorectal cancer cells, LoVo and HT29, and their doxorubicin-resistant counterparts, LoVo/Dx and HT29/Dx, respectively.

The paper fits within the scope of IJMS and presents an original topic that is certainly of interest for the readers of this journal. However, some points need to be improved in the article before being publishable:

  1. In the introduction section, the Authors should remove the results of the study;
  2. The Authors should remove from paragraph “2.1. Identification of Compound” the methods of extraction of the compounds, which, instead, must be moved in the methods;
  3. In the paragraph “2.2. Cytotoxic Activity of β-Cyclocitral and its Derivative TMPE against Colon Cancer Cells” the Authors stated that TMPE inhibited growth of resistant cells in a “concentration dependent manner”. However, looking at figure 2B, this effect is not observable.
  4. I suggest the Authors to revise the statistic of figure 2B, because, in my opinion, it is wrong.
  5. Throughout the text, Authors must pay attention to typographical errors (for example, when writing µM concentrations and Greek letters in general);
  6. Since the study has been performed in an in vitro model, please replace “dose” with “concentration” (for example page 4 lines 138 and 145);
  7. Why did the Authors evaluate the accumulation of Rhodamine 123 only in HT29/Dx cells and not also in LoVo/Dx? Both overexpress ABCB1 protein, therefore Authors must evaluate this parameter in both cell lines.
  8. The figure 5 lacks of the statistical significance; moreover, Authors should have used a positive control like verapamil to compare the effect of a known MDR protein inhibitor to that of TMPE.
  9. In the paragraph “4.5. Western Blot Analysis”, the Authors stated that densitometric analysis of the bands was performed using ImageJ software. However, in figure 7, the Authors did not show the densitometry. Please, add it;
  10. All results are written approximately, therefore Authors are invited to give more details;
  11. In the “4.10. Statistical Analysis”, the Authors stated that “the values of measured parameters were presented as the means from three independent experiments ± standard errors (SEM)”, but in all results paragraph, they said that the results are expressed as mean ± S.D. of three experiments. They should resolve this discrepancy;
  12. Figures 8 and 9 have panel letters in a misleading position. Authors should put them before the relative image. Moreover, Authors should specify in the legend the software through which they obtained the abovementioned images;
  13. Please revise the English language throughout the article.

Reviewer 2 Report

In experimental part of manuscript lack of key experiments and appropriate controls to interpret results correctly and firmly. The authors should provide the explanation of the mechanism of TMPE activity in doxorubicin resistance in CRC.

 Major concerns:

  1. Why the authors select for experiments two particular colon cancer lines, i.e. LoVo and HT29. They have different origin and genetic background please comment that fact in manuscript in the light of doxorubicin resistance and its reversal.
  2. Doxorubicin is not currently used in CRC therapy please comment that fact in manuscript. Why CRC lines were selected as an experimental model ? Doxorubicin resistance is not to be adequate issue for further translational research in CRC chemotherapy.
  3. In Fig.2 provide the doubling time analysis of both cells lines. This analysis is necessary to correctly interpret the cytotoxicity analysis results.
  4. ,,Both, LoVo/Dx and HT29/Dx cell lines are characterized by overexpression of ABCB1 protein’’, please provide the Western blot of ABCB1 expression in both lines used in experiments. What about THE other doxorubicin transporters, ABCC1 and ABCG2, provide Western blot showing their expression in both cells lines. Comment the involvement of those transporters in doxorubicin resistance.
  5. TMPE inhibited transport activity of ABCB1 protein without affecting its expression level. Then, what is the mechanism of this inhibition ? How about the subcellular localization of ABCB1 under TMPE treatment ? Provide the analysis of ABCB1 cellular localization.
  6. Analyze how TMPE affects ATP-ase activity of ABCB1. This analysis and subcellular localization of ABCB1 analysis ( see p.5) will show how TMPE impact ABCB1 protein activity.
  7. Analysis of TMPE physicochemical parameters suggested it was not likely to be transported by ABCB1. Again, what is the mechanism of this inhibition? How TMPE affects ABCB1 activity other than by its inhibition or substrate competition. Please provide explanation and results.
  8. In Fig.7 provide quantitative analysis of presented WB, Ponceau staning presents only protein pattern after transfer, do not specify protein bands on it, only Ab staining identify particular protein, correct the figure and remove the ponceau staining or relocate it to supplementary data.
  9. How TMPE may affect other protein or signaling pathway in cells. Please provide analysis of key pathway related to cell survival in cells treated with TMPE.
  10. In manuscript lack of experiments that establish role of TMPE in dox resistance. Please provide the mechanism of TMPE activity to strengthen presented observation.

Round 2

Reviewer 1 Report

Author addressed my suggestions thoroughly.

Author Response

The reply was not required.

Reviewer 2 Report

Please revise/update the knowledge about colon cancer biology, e.g. current classification, EMT vrs. metastatic spread, 

Author Response

The Introduction and Discussion parts of the manuscript were changed to introduce colon cancer biology and EMT information. Thank you for your review.